# Imidazolyl Ethanamide Pentandioic Acid (IEPA) as Potential Radical Scavenger during Tumor Therapy in Human Hematopoietic Stem Cells

**DOI:** 10.3390/molecules28052008

**Published:** 2023-02-21

**Authors:** Lucas C. Pfau, Annegret Glasow, Clemens Seidel, Ina Patties

**Affiliations:** Department of Radiation Oncology, University of Leipzig, 04103 Leipzig, Germany

**Keywords:** imidazolyl ethanamide pentandioic acid, CD34^+^ human stem and progenitor cells, glioblastoma, head and neck squamous cell carcinoma, reactive oxygen species, radiotherapy, radioprotector, chemotherapy, radical scavenger

## Abstract

Radiochemotherapy-associated leuco- or thrombocytopenia is a common complication, e.g., in head and neck cancer (HNSCC) and glioblastoma (GBM) patients, often compromising treatments and outcomes. Currently, no sufficient prophylaxis for hematological toxicities is available. The antiviral compound imidazolyl ethanamide pentandioic acid (IEPA) has been shown to induce maturation and differentiation of hematopoietic stem and progenitor cells (HSPCs), resulting in reduced chemotherapy-associated cytopenia. In order for it to be a potential prophylaxis for radiochemotherapy-related hematologic toxicity in cancer patients, the tumor-protective effects of IEPA should be precluded. In this study, we investigated the combinatorial effects of IEPA with radio- and/or chemotherapy in human HNSCC and GBM tumor cell lines and HSPCs. Treatment with IEPA was followed by irradiation (IR) or chemotherapy (ChT; cisplatin, CIS; lomustine, CCNU; temozolomide, TMZ). Metabolic activity, apoptosis, proliferation, reactive oxygen species (ROS) induction, long-term survival, differentiation capacity, cytokine release, and DNA double-strand breaks (DSBs) were measured. In tumor cells, IEPA dose-dependently diminished IR-induced ROS induction but did not affect the IR-induced changes in metabolic activity, proliferation, apoptosis, or cytokine release. In addition, IEPA showed no protective effect on the long-term survival of tumor cells after radio- or chemotherapy. In HSPCs, IEPA alone slightly enhanced CFU-GEMM and CFU-GM colony counts (2/2 donors). The IR- or ChT-induced decline of early progenitors could not be reversed by IEPA. Our data indicate that IEPA is a potential candidate for the prevention of hematologic toxicity in cancer treatment without affecting therapeutic benefits.

## 1. Introduction

Since the discovery of radioactivity and ionizing radiation and their first medical applications at the end of the nineteenth century, the understanding of their underlying physics and radiobiology has increased dramatically. This allows for very precise application of IR in modern radiotherapy (RT). By comparison, the means of protecting the human body from the adverse effects of IR remain limited. Possible applications in radiation oncology, space flight, and protecting the public in cases of nuclear accidents or war have motivated extensive research into pharmaceutical agents for prophylaxis (radioprotectors), treatment soon after exposure (radiomitigators), and therapy for radiation injury [1,2]. There are agents approved by the FDA, such as amifostine, palifermin, G-CSF, and GM-CSF, and several promising compounds are under development [2,3,4]. Due to limitations regarding adverse effects, the spectrum of application, and effectiveness, there is still an urgent need for a potent and well-tolerated drug [2,3].

Even in a controlled clinical setting, the management of the adverse effects of RT is one of the most challenging tasks for any radiation oncologist, as these effects are often a therapy-limiting factor. This is especially true in concurrent chemoradiotherapy (ChRT) [5], which is utilized, e.g., in therapy for head and neck squamous cell carcinoma (HNSCC) and glioblastoma (GBM) [6,7,8]. Importantly, acute adverse effects include thrombocytopenia, neutropenia, and anemia, and about 40% of HNSCC and GBM patients are affected [9,10]. RT may induce a wide range of short-term (associated with acute cell damage and inflammation) and long-term (characterized by repair and remodeling) adverse effects, including myelosuppression [8,10,11,12]. In the treatment of HNSCC and GBM, however, myelosuppressive effects are caused to a greater extent by concomitant ChT and are leading causes of dose reduction and treatment delays [5,11,12,13]. Cisplatin (CIS) is used in HNSCC therapy, but in addition, alkylating agents such as temozolomide (TMZ) and lomustine (CCNU) for GBM treatment are well known for their bone marrow toxicity [10,11,12,13,14]. Bone marrow toxicity is not only a dangerous acute condition for patients prone to infections or bleeding, it also carries the risk of reduced total chemotherapy dose, which is known to be potentially detrimental to oncologic outcomes, e.g., in patients with head and neck cancer [15,16].

While bone marrow damage induced by ChT or RT is of different origin, both result in the impairment of CD34^+^ hematopoietic stem and progenitor cells (HSPCs) [5]. This heterogeneous group of cells is responsible for stable hematopoiesis. For this vital role, it depends on highly specific conditions and is very sensitive to changes in its bone marrow microenvironment [17,18]. Recombinant growth factors such as G-CSF (e.g., filgrastim) or thrombopoietin (TPO; e.g., romiplostim) are already clinically used to boost impaired granulopoiesis and thrombopoiesis [19,20]. This, however, does not protect vulnerable CD34^+^ HSPCs from the effects of RT or ChT in the first place but contributes to an exhaustion of the bone marrow in the long term [21]. With myelosuppression being relevant both in radiation oncology and in the pathophysiology of acute radiation syndrome (as hematopoietic subsyndrome), these cells are of special interest in the development of radioprotectors [3].

Imidazolyl ethanamide pentandioic acid (IEPA) is an orally bioavailable pseudopeptide of low molecular weight which is used in Russia as an antiviral agent in the treatment of influenza and other respiratory viruses [22,23]. The mode of action has not been determined completely, but intriguingly, it shows activity against infections by multiple phylogenetically different viruses in mice. Specifically, IEPA shows cytoprotective activity and prevents cell death [23]. Additionally, IEPA has previously been shown to induce differentiation and maturation in mouse bone marrow HSPCs after treatment with cyclophosphamide and to thereby reduce ChT-induced neutropenia in the peripheral blood [22]. A small clinical trial in Hodgkin lymphoma patients [24] and animal studies of irradiated rabbits [25] indicate that IEPA may also have the potential to reduce myelosuppression and exert hematoprotective effects when combined with ChT or RT.

For a potential clinical application, it is important to verify if the IEPA-induced cellular protection regarding hematopoietic cells extends to tumor cells impairing the antitumor effect of ChT/RT. Therefore, in this study we investigated IEPA’s mode of action and effects on multiple levels of cellular function in HNSCC (FaDu) and GBM (A172) tumor cells as well as in CD34^+^ HSPCs. In order to analyze the potential radio-/chemoprotective effects of IEPA, it was applied 1 h before RT and ChT with CIS, TMZ, or CCNU.

In tumor cell lines, we assessed the ROS induction, metabolic activity, cell proliferation, apoptosis induction, cytokine production, and clonogenic survival to determine whether IEPA has protective properties on human tumor cells that render it unsafe for use in clinical trials in a radio-oncological setting.

In HSPCs isolated from the umbilical cord blood of several donors, we tested for the effects of IEPA on cytokine release, differentiation behavior, apoptosis induction, proliferation, and induction of DSBs to further elucidate its mechanism of action on these cells.

## 2. Results

### 2.1. Metabolic Activity: Dose-Finding Experiments with IEPA, IR, and ChT in Tumor Cells

To examine IEPA effects on metabolic activity, we first determined half-maximal inhibitory IR doses and drug concentrations (ID_50_/IC_50_) representing a benchmark for appropriate damage induction throughout the following experiments (data not shown). Based on clinical practice, the chemotherapeutic agents TMZ (fractionated treatment) and CCNU (single treatment) were applied to glioblastoma (A172) cells and CIS (single treatment) to head and neck cancer (FaDu) cells. The examined ID_50_/IC_50_ values and equivalent irradiation doses according to the linear–quadratic model are presented in Table 1.

At high doses, IEPA reduced the metabolic activity slightly in both tumor entities, by 9.9 ± 1.8 and 7.9 ± 4.5% at 20 and 100 µM, respectively (joint analysis, n = 5, *p* ≤ 0.05). In cells treated with IR (ID_50_) and ChT (IC_50_; CIS, TMZ, CCNU), IEPA did not influence the induced reduction in metabolic activity in any concentration (joint analysis, n = 5) (Figure 1).

### 2.2. Effects of IEPA and IR on Proliferation in Tumor Cells and CD34^+^ HSPCs

To study the effects of IEPA on cell proliferation, a BrdU (tumor cells) or EdU incorporation assay (CD34^+^ HSPCs) was performed 48 and 72 h after single-dose IR (9 Gy).

Tumor cells showed a time-dependent reduction in proliferation activity to 65.1 ± 11.8% (FaDu) and to 41.8 ± 8.3% (A172) at 72 h after IR (n = 2, *p* ≤ 0.05 in FaDu; *p* ≤ 0.001 in A172; Figure 2A). IEPA alone increased the proliferation only slightly in A172 cells, to 116.3 ± 4.3% (10 µM) and 118.5 ± 4.6% (100 µM) 72 h after treatment (n = 2, *p* ≤ 0.01). IEPA given 1 h prior to IR had no effect on IR-induced proliferation decline, though a tendency for an additional dose-dependent reduction was seen in both tumor cell lines.

In CD34^+^ HSPCs, proliferating (EdU^pos^) cells were reduced from 30.8 ± 2.0% to 9.2 ± 0.9% (48 h) and from 40.7 ± 2.9% to 15.9 ± 0.4% (72 h) after IR (Figure 2B,C). The gradual differentiation of CD34^+^ HSPCs could be excluded as a cause, as the level of CD34^+^ cells remained stable at 75.8 ± 0.8%. Regarding proliferation and differentiation, neither an effect of IEPA (1–100 µM) alone nor an effect of IEPA combined with IR could be identified.

### 2.3. Evaluation of Cell Death after IEPA and IR or ChT Treatment in Tumor Cells and CD34^+^ HSPCs

To examine the effect of IEPA on apoptosis and necrosis, Annexin-V- and PI-positive cells were evaluated 72 h after treatment.

The apoptotic cell fraction of tumor cells ranged from 4.1 (single dose: 9 Gy) to 69.1% (4 × 5.9 Gy) in A172 cells and from 9.8 (4 × 1.8 Gy) to 74.6% (4 × 5.9 Gy) in FaDu cells (Figure 3B). IEPA alone or combined with IR had no additional effects. The number of necrotic cells in FaDu ranged from 0.20 ± 0.03% (the joint value of all groups treated with sham-IR with/without IEPA) to 0.99 ± 0.16% (the joint value of all groups treated with IR with/without IEPA), and in A172 cells it ranged from 1.47 ± 0.32 (sham-IR) to 3.75 ± 0.21% (IR). No changes due to IEPA could be observed.

CD34^+^ HSPCs of two donors were tested in independent experiments (n = 1) showing different baseline levels of apoptotic cells (Figure 3C). Apoptosis was induced in both donors by IR (3.2 Gy) or treatment with CIS (1 µM), TMZ (40 µM), and CCNU (60 µM). IEPA (100 µM) showed no effect on the levels of apoptotic CD34^+^ HSPCs.

### 2.4. Effects of IEPA and IR or ChT Treatment on Long-Term Clonogenic Survival of Tumor Cells

To examine the long-term effects of IEPA combined with IR (single-dose: 3.2–9.0 Gy) or chemotherapeutic treatment (FaDu: 1 µM CIS; A172: 60 µM CCNU, 10 µM TMZ daily and single-dose) on the clonogenic survival of tumor cells, we performed clonogenic assays. After IR, a dose-dependent reduction of the surviving fractions to 1.1 × 10^−2^ ± 2.1 × 10^−3^ (FaDu) and 2.8 × 10^−2^ ± 1.9 × 10^−3^ (A172) at 9 Gy was observed (Figure 4B,C). The tested cytostatic agents decreased the surviving fractions to 6.4 × 10^−1^ ± 7.5 × 10^−2^ (CIS), 4.4 × 10^−2^ ± 9.1 × 10^−3^ (CCNU), 2.5 × 10^−2^ ± 8.8 × 10^−3^ (TMZ daily, 4 fractions), and 1.1 × 10^−1^ ± 8.9 × 10^−3^ (TMZ, single-dose). IEPA alone and in combination showed no protective effect on the long-term survival of clonogenic tumor cells. In A172 cells treated with CCNU and TMZ (daily), 100 µM IEPA seemed to further decrease the clonogenic survival.

### 2.5. Effect of IEPA on IR- or ChT-Induced Reactive Oxygen Species (ROS) in Tumor Cells and CD34^+^ HSPCs

To examine the effect of IEPA on the amount of IR- or ChT-induced ROS, we performed a DCFDA assay after the treatment of tumor cells and CD34^+^ HSPCs. Immediately after 5.5 Gy single-dose IR, we observed 5-fold, 3.7-fold, and 6.5-fold increases in intracellular ROS levels in FaDu, A172 (n = 3, *p* ≤ 0.001), and CD34^+^ HSPCs (n = 1), respectively (Figure 5A).

IEPA alone had no effect, but it reduced the IR-induced ROS significantly in a dose-dependent manner by up to 30 ± 16% at 100 µM (n = 3, *p* ≤ 0.001) in both tumor cell lines. We also observed this effect in CD34^+^ HSPCs, albeit to a lesser extent (15% reduction with 100 µM IEPA; n = 1).

At 48 h after IR, the amount of ROS was still enhanced 1.6-fold (FaDu) and 2.2-fold (A172) but not affected by IEPA anymore (Figure 5B, A172).

The tested chemotherapeutic drugs (TMZ single-dose: 20/40 µM, CCNU: 30/60 µM, CIS: 0.5/1 µM) showed no increased ROS levels up to 5 h after application and therefore no effect of IEPA could be observed (data not shown). However, at 48 h, 1.3-fold (FaDu) and 1.2-fold (A172) increases in ROS could be detected after a very high concentration of CIS (20 µM) but without IEPA having an effect (Figure 5C, FaDu).

### 2.6. Effects of IEPA on Cytokine Release after IR of Tumor Cells and CD34^+^ HSPCs

The release of cytokines IL-1β, IL-6, IL-8, and TNF-α was analyzed in the supernatants of tumor cells (24 h) and CD34^+^ HSCPs (24, 48, 72 h) after IR (single-dose: 9 Gy) in order to investigate the influence of IEPA on inflammatory responses.

In tumor cells only IL-6 and IL-8 could be measured after 24 h. FaDu cells showed higher cytokine levels compared with A172 cells (IL-6: FaDu: 3.38 pg/mL, A172: 0.14 pg/mL; IL-8: FaDu: 5.96 pg/mL, A172: 0.17 pg/mL) (Figure 6A). However, IR did not increase these cytokines further. IEPA seemed to exhibit a biphasic effect for both cytokines in FaDu with increased secretion at 100 µM. In A172, no effect of IEPA could be observed.

In CD34^+^ HSPCs, only the release of IL-8 was measurable. In contrast to tumor cells, IR induced a 10- to 19-fold increase compared with control cells (from 5.3, 3.1, and 6.6 pg/mL to 61.5, 57.7, and 62.7 pg/mL, respectively, after 24, 48, and 72 h; Figure 6B). IEPA did not show an effect on IL-8 release in CD34^+^ HPSCs.

### 2.7. Effects of IEPA and IR or ChT Treatment on the Differentiation Behavior of CD34^+^ HSCPs after Treatment

To examine the effect of IEPA (100 µM) on the differentiation behavior of CD34^+^ HSPCs, colony-forming unit cell (MethoCult™) assays were performed and CD34^+^ HSPC-derived colonies were counted (Figure 7A,B).

In general, CD34^+^ HSPCs showed less sensitivity towards cytostatic agents and more radiation sensitivity compared with previous experiments. Thus, we were required to double concentrations for TMZ (40 µM, single-dose), CCNU (60 µM), and CIS (1 µM) and decrease IR doses (single-dose: 3.2 Gy, fractionated: 4 × 1 Gy).

In both donors, a trend towards decreases in total colony count to 15.4 ± 4.8% (3.2 Gy), 15.2 ± 4.1% (4 × 1 Gy), 63.9 ± 7.5% (TMZ), 74.1 ± 0.7% (CCNU), and 65.1 ± 2.5% (CIS) compared with control was seen. IEPA alone did not influence the total colony count but induced increases in early progenitor cells in both donors by 30 ± 9.1% (CFU-GEMM (granulocyte-erythroid-monocyte-megakaryocyte)) and 28.1 ± 15.5% (CFU-GM (granulocyte-monocyte)), respectively.

IEPA combined with CCNU reduced the CFU-GM counts in both donors from 93.3 ± 11% to 78.7 ± 8.1%. In combination with TMZ, IEPA reduced the CFU-G (granulocyte) counts slightly from 58.7 ± 5.8% to 50.8 ± 7.4%. IEPA combined with CIS (53.9 ± 15.5% to 27.4 ± 3.7%) or fractionated IR (38.7 ± 12.3% to 16.6 ± 6.0%) reduced the CFU-M (monocyte) counts slightly. However, it should be mentioned that only a small number of CFU-M could be counted (0.42 ± 0.05 CFU-M/100 CD34^+^ HSPCs).

### 2.8. Effects of IEPA on Initialization and Repair Kinetic of IR-Induced DNA Double-Strand Breaks

The impact of IEPA on IR-induced DNA double-strand breaks (DSBs) in CD34^+^ HSPCs was assessed through the analysis of γH2AX foci at 0.5, 4, and 24 h after treatment.

IR (3.2 Gy) induced an increase in γH2AX foci per nucleus from 2.7 ± 0.5 in the sham-irradiated control to 12.3 ± 1.4 after 0.5 h (Figure 8A). Time-dependent decreases in IR-induced foci to initial levels were observed at 4 h (9.1 ± 1.2) and at 24 h (3.5 ± 0.4), indicating DSB repair. IEPA showed no effect in preventing the initial formation of DSBs and did not affect the repair process of DNA damage.

## 3. Discussion

To assess the suitability of IEPA as a potential radiation protector for hematopoietic stem cells in a radiochemotherapeutic setting, the question of whether IEPA would also protect tumor cells from radiation needed to be answered.

Therefore, we investigated the effect of IEPA in combination with IR and tumor-specific chemotherapeutic agents in two tumor entities, represented by the human cell lines FaDu (HNSCC) and A172 (glioblastoma). In order to gain more insight into its acting mechanism, we also compared its activity with CD34^+^ HSCPs isolated from human cord blood.

Detailed analyses of important cellular functions such as metabolic activity, apoptosis, proliferation, DNA damage repair, and clonogenic survival, and of stress response mechanisms such as ROS and cytokine release, were performed to evaluate the effects of IEPA on radiochemotherapy-induced alterations.

First, the ID_50_/IC_50_ values were calculated for IR and chemotherapeutic treatments, measuring the metabolic activity in tumor cells. The determined IC_50_ concentrations for TMZ and CIS are given below and for CCNU were 10-fold higher than the reported maximum plasma concentrations (CCNU 3.2 µM [26], TMZ 50 µM [27], and CIS 10 µM [28]). IEPA has been applied in clinical trials for chemotherapy-induced neutropenia (100 mg/day; MyeloConcept trial) [29,30], as a protector of myelodepression in Hodgkin’s disease [24] (90–100 mg/day), and also in children for the treatment of respiratory viral infections (30 mg/day) [31]. Human plasma concentrations of 579 ng/mL (2.6 µM), measured 2 h after intake of 90 mg IEPA [32], are covered by the dose range of our experiments.

Tumor cell lines: The application of IEPA alone had no significant effect on apoptosis, clonogenic survival, ROS, or cytokine production in either tumor cell line over a broad concentration range (0.1–100 µM) (Figure 3B, Figure 4B,C, Figure 5 and Figure 6A). Only a minor impact on the metabolic activity and proliferation of A172 cells was seen at high doses (Figure 1 and Figure 2A). Although IEPA alone tended to reduce the metabolic activity slightly in A172 cells, it did not alter the effect of IR or ChT (Figure 1). Similarly, despite slightly enhancing the tumor cell proliferation in A172 cells, no tumor-protecting activity could be derived, since this effect was no longer present in combination with IR (Figure 2A).

Clonogenic survival was analyzed to evaluate long-term effects on cancer cell reproductivity, an important indicator for tumor regrowth [33]. IR, CCNU, TMZ, and CIS treatments caused, as expected from their clinical applications, a decline in clonogenicity in the respective cell lines [34,35]. Most importantly, IEPA, either applied alone or in combination with IR or ChT, did not show a protective effect on clonogenic survival in the tested cell lines (Figure 4B,C).

IR inflicts cellular damage partly through the generation of ROS [36,37]. Accordingly, we showed an elevation of ROS immediately following IR (Figure 5A), which is in line with previous findings [38,39,40]. The continuing ROS elevation 48 h after IR and CIS in A172 and FaDu cells (Figure 5B,C) may be a result of IR-induced mitochondrial dysfunction [41]. Cisplatin has already been shown to induce mitochondrial ROS in other tumor cell lines [42], which can also be confirmed here. IEPA was able to reduce the initial IR-induced ROS concentration on both tumor cell lines to some extent, although these effects did not change the viability of irradiated tumor cells. Tumor survival probably does not depend on ROS induction to a major extent. After decades of research, it is still not fully clear whether ROS have protective or damaging effects on tumor tissue or how antioxidant modulation may influence the efficacy of radiotherapy or chemotherapy [43].

IR is known to exert acute and chronic inflammatory effects, e.g., by involving redox-sensitive transcription factors such as NF-κB, leading to enhanced secretion of cytokines [44,45]. Proinflammatory cytokines have been reported to be secreted in FaDu cells (IL-6, IL-8, and TNF-α) as well as in A172 cells (IL-1β, IL-6, and IL-8) [46,47]. The combination of IR (9 Gy) and IEPA as well as single agents, however, did not significantly alter cytokine release in either tumor cell line (Figure 6A), which points towards the deregulated cytokine signaling commonly found in cancer cells [44].

CD34^+^ HSPCs: Primary cord blood-derived CD34^+^ HSPCs showed high radio- and chemosensitivity towards apoptosis induction (Figure 3C). Compared with tumor cells, proliferation inhibition was also stronger after IR (Figure 2B), similar to the high radiosensitivity described in bone-marrow-derived HSPCs [48]. The administration of IEPA alone up to 100 µM had no adverse effects on HSPC cell death and proliferation, which is consistent with a broad clinical and experimental record of IEPA as a substance with low toxicity and low risk of overdose [23,30]. However, no rescue effect of IEPA was found in IR-induced responses, despite its myeloprotective properties reported in vivo [25]. Divergent results might be explained by short- vs. long-term survival effects, which can consequently lead us to conduct clonogenic long-term survival analyses.

Keeping in mind that CD34^+^ HSPCs are a diverse group of cells containing long-term (LT) and short-term (ST) repopulating hematopoietic stem cells (HSCs) as well as progenitor cells, which gradually lose their potential for self-renewal in the process of differentiation, we needed to detect and quantify different progenitor populations (Figure 7). The higher radiosensitivity of CD34^+^ HSPCs compared with tumor cells forced us to lower the IR dose (3.2 Gy instead of 9 Gy) to obtain evaluable results in all subpopulations. CD34^+^ HSCs and quiescent LT-HSCs are particularly vulnerable [49], are more prone to undergoing apoptosis instead of DNA damage repair, and show impaired clonogenic potential [50,51,52]. On the other hand, CD34^+^ HSPCs showed less sensitivity towards cytostatic agents and relatively high concentrations of IEPA (100 µM) were needed to be applied for significant effects. Using a colony-forming unit cell assay, we could demonstrate an increase in early HSPCs (CFU-GEMM) and CFU-GM colonies using IEPA alone in both donors, indicating that IEPA might be able to enhance these cell populations in a preventive setting. Regarding the implications of our findings on CD34^+^ HSPC differentiation and functionality, we must consider that the hematopoietic BM niche that HSPCs reside in is a highly complex microenvironment that regulates their function and allows for controlled hematopoiesis. A balance of LT/ST-HSC quiescence, proliferation, and differentiation is provided by mesenchymal, endothelial, and neuronal cells secreting regulatory cytokines and chemokines [18,48,53]. Despite the use of hematopoietic cytokines (StemSpan^TM^ CC110 containing Flt-3 Ligand, stem cell factor (SCF), and TPO in our experiments, it was not possible to properly simulate all BM features such as cytokines, cellular influences, or low oxygen levels in an in vitro single-cell suspension. Thus, the possible effects of IR and IEPA on the microenvironment and interactions with other BM cell types which finally affect the survival and differentiation of HSPCs could not be examined. Moreover, the umbilical cord blood-derived CD34^+^ HSCPs that were used might not behave like BM-derived HSCs or hematopoietic progenitor cells (HPCs) [54].

The role of intracellular ROS in CD34^+^ HSPCs is the subject of extensive research. Low levels of ROS in LT-HSCs seem to be another important balancing factor [48,53], partly achieved by hypoxic conditions and low blood perfusion in the HSC niche and surrounding BM cells but also by the HSC metabolic profile [17,18,55]. LT-HSCs primarily utilize anaerobic glycolysis as a means of ATP synthesis, thereby reducing mitochondrial ROS production [56]. The need for the exact regulation of oxidative stress might play a role in the high radiosensitivity of HSPCs [48,49,50,57], which exhibit higher levels of persistent ROS generation after IR. Accordingly, we found a strong IR-induced elevation of ROS in CD34^+^ HSPCs (Figure 5A). IEPA was only slightly able to scavenge IR-induced ROS, with no influence on radiosensitivity regarding proliferation, apoptosis, cytokine release, or DNA repair (Figure 2B, Figure 3C, Figure 6B and Figure 8A, respectively). As mentioned previously, interpretation of these results is limited by the artificial conditions of the in vitro culture, especially regarding the higher oxygen partial pressure but also due to the varying sensitivities of HSPCs depending on their origin. Therefore, the possibility of underestimation of IEPA’s effects on BM HSPCs cannot be excluded.

Partially linked to ROS-induced stress [58], radiotherapy is often limited by the induction of inflammation along with the autocrine/paracrine release of HSC-proliferation-inducing cytokines such as IL1-ß, IL-6, IL-8, or TNF-α [59,60,61,62]. IEPA itself has been described as inducing TNFα in THP1 monocytes [63]. In our cord blood-derived CD34^+^ HSPCs, only IL-8 protein could be detected (Figure 6B), which is in line with previous findings that show either lower or no protein expression of IL-1ß, TNF-α, or IL-6 compared with IL-8 in HSPCs [64,65]. This was strongly induced by IR, possibly via the ROS/TNFα pathway, and could contribute to the mobilization of stem cells as part of a stress response mechanism [64,65,66]. Again, IEPA had no effect on IL-8 secretion.

## 4. Materials and Methods

### 4.1. Reagents

Cisplatin ((SP-4-2)-diamminedichloridoplatinum(II); CIS), lomustine (N-(2-Chloroethyl)-N’-cyclohexyl-N-nitrosourea; CCNU), and temozolomide (4-methyl-5-oxo-2,3,4,6,8-pentazabicyclo[4.3.0]nona-2,7,9-triene-9-carboxamide; TMZ) were purchased from Sigma-Aldrich (St. Louis, MO, USA). Imidazolyl ethanamide pentandioic acid (5-[2-(1*H*-imidazol-5-yl)ethylamino]-5-oxopentanoic acid; IEPA; for chemical structure see Figure 9) was kindly provided by MyeloTherapeutics (Berlin, Germany). Stock solutions were prepared as follows: 30 mM CIS (molecular weight (MW) 300 g/mol) in 33% dimethyl sulfoxide (DMSO); 100 mM CCNU (MW 249.69 g/mol) in 5% ethanol; 100 mM TMZ (MW 194 g/mol) in 10% DMSO; and 100 mM IEPA (MW 225.25 g/mol) in phosphate-buffered saline (PBS; Lonza, Basel, Switzerland). Aliquots were stored at −20 °C and further working solutions were prepared with assay-specific culture medium (see cell culture and labeling of CD34^+^ cells) or PBS. Appropriate solvent controls were implemented.

### 4.2. Cell Culture of Tumor Cell Lines

The human glioblastoma cell line A172 was purchased from ATCC (Manassas, VA, USA). Cells were grown in Dulbecco’s modified Eagle’s medium (DMEM; Lonza, Basel, Switzerland) supplemented with 10% fetal bovine serum (FBS; Sigma-Aldrich, St. Louis, MO, USA), 100 U/mL penicillin, and 100 µg/mL streptomycin (Lonza, Basel, Switzerland). The human pharynx squamous cell carcinoma cell line FaDu was kindly provided by TU Dresden, Germany (Prof. Krause, Department of Radiotherapy and Radiation Oncology). Cells were cultured in Eagle’s minimum essential medium (EMEM; Lonza, Basel, Switzerland) containing 10% FBS, 2 mM L-glutamine (Sigma-Aldrich, St. Louis, MO, USA), 100 U/mL penicillin, and 100 µg/mL streptomycin. Both cell lines were cultivated at 37 °C and 5% CO_2_ and passaged twice a week by trypsinization to a maximum of 50 passages.

### 4.3. Isolation and Culture of Human CD34^+^ Cells

Human umbilical cord blood mononuclear cells (hUCB-MNCs) were purchased from Vita34 AG (Leipzig, Germany) and stored in liquid nitrogen until usage. Cells were thawed carefully using CD34^+^ isolation buffer (PBS, 0.5% bovine serum albumin (BSA; Serva Electrophoresis GmbH, Heidelberg, Germany), 2 mM EDTA (Carl Roth GmbH & Co. KG, Karlsruhe, Germany)) supplemented with 10% FBS at 4 °C. Cells were then centrifuged (300× *g*, 10 min, 4 °C) and living cells were counted using trypan blue (Sigma-Aldrich, St. Louis, MO, USA) exclusion method. To prevent clumping, cells were resuspended in PBS containing 0.5% BSA, 2 mM MgCl_2_ (Carl Roth GmbH & Co. KG, Karlsruhe, Germany), and 100 U/mL deoxyribonuclease 1 (Sigma-Aldrich, St. Louis, MO, USA). After incubation for 20 min at 37 °C, CD34^+^ cells were isolated using the CD34 MicroBead Kit, human (order no. 130-046-702, Miltenyi Biotec, Bergisch Gladbach, Germany) according to manufacturers’ instructions. Briefly, cells were labeled with anti-CD34 magnetic beads, washed with CD34^+^ isolation buffer, and applied to an MS MACS column (Miltenyi Biotec, Bergisch Gladbach, Germany) placed in MiniMACS™ separator. Columns were washed four times with CD34^+^ isolation buffer followed by elution of CD34^+^ cells using a plunger in the absence of the magnetic field. To increase the purity of the CD34^+^ cells, the eluted fraction was applied to a second column and magnetic separation steps were repeated. After the living cells were counted, they were stained with APC mouse antihuman CD34 (Clone 8G12; Becton Dickinson, Franklin Lakes, NJ, USA) for 30 min at room temperature. An appropriate isotype control (APC Mouse IgG_1_; Clone X40; Beckton Dickinson, Franklin Lakes, NJ, USA) was implemented and the purity of the isolated cells was assessed with flow cytometry (BD Accuri C6 Plus, Becton Dickinson, Franklin Lakes, NJ, USA). Cells were grown in StemSpan^TM^ Serum-free Expansion Medium (SFEM, Stemcell Technologies, Vancouver, Canada) supplemented with StemSpan^TM^ CC110 (Stemcell Technologies, Vancouver, BC, Canada) in a 1:100 dilution for subsequent assays if not otherwise noted.

### 4.4. Ionizing Radiation (IR)

An X-ray machine (Xstrahl 200, Xstrahl GmbH, Ratingen, Germany) was used at 150 kV and a dose rate between 1.06 and 1.69 Gy/min depending on the target was used. In combinatorial treatments, IR was conducted 1 h after IEPA supplementation. Sham-irradiation was performed at equal conditions.

Using the linear–quadratic model [67] with α/β = 10 [68], the biologically effective dose (BED) of fractionated irradiation was calculated (Equation (1)) and translated to the equivalent single dose. A sample calculation for 4 × 1.8 Gy is shown in Equation (2). The ID[BED]_50_ was used for the calculation of multiples of ID_50_ values in both fractionated and single-dose irradiation regimens (Table 1).

Equation for BED calculation:(1)$$BED=n\times d\left(1+\frac{d}{\left[\alpha /\beta \right]}\right)$$

*BED*… Biologically effective dose 

*n*… Number of fractions 

*d*… Dose per fraction

*α*/*β*… Tissue-specific factor [68]

EXAMPLE

Objective: *BED*

*n* = 4, *α*/*β* = 10, *d* = 1.8 Gy
(2)$$\begin{array}{l} BED=4\times 1.8\;Gy\left(1+\frac{1.8\;Gy}{10Gy}\right)\\ BED=8.5\;Gy \end{array}$$

Objective: equivalent single dose *d*

*BED* = 8.5 Gy, *n* = 4, *α*/*β* = 10
8.5 Gy=d(1+d10 Gy)8.5 Gy=d+d210 Gy85 Gy=d2+10 Gy×d0=d2+10 Gy×d−85 Gy0=d2+2×5 Gy×d+52Gy−(85 Gy+52Gy)0=d2+2×5 Gy×d+25 Gy−110 Gy110 Gy=d2+2×5 Gy×d+25 Gy110 Gy=(d+5)2 Gy110  Gy=d+5 Gyd=110  Gy−5 Gyd=5.5 Gy×10 Gy 1st binominal formula 


The equivalent single dose to 4 × 1.8 Gy fractionated irradiation with *α*/*β* = 10 Gy is 5.5 Gy.

### 4.5. Treatment Schedule

Tumor cells were allowed to adhere 24 h prior to treatment. If not otherwise noted, IEPA was applied 1 h prior to treatment with cytostatic agents or irradiation. Based on the clinical practice, chemotherapeutic agents TMZ (single or fractionated treatment) and CCNU (single treatment) were applied to glioblastoma (A172) and CIS (single treatment) to head and neck cancer (FaDu) cells. IR (single-dose or fractionated) was conducted immediately after drug application. If not otherwise noted, fractionation (TMZ and IR) was performed daily for 4 consecutive days with medium renewal before IEPA administration. The concentrations of cytostatic agents and IR doses in combination experiments were adapted from preliminary metabolic activity measurements in tumor cells (Figure 1) and correspond to the respective half-maximal inhibitory doses/concentrations (ID_50_/IC_50_) or multiples of these (Table 1).

### 4.6. Metabolic Activity (WST-1)

The treatment effects on metabolic activity of tumor cells were measured using the tetrazolium-salt-based reagent WST-1 (Roche, Basel, Switzerland). Cells were seeded in 24-well plates (3000 cells/1 mL culture medium per well) and treated with the following doses/concentrations: IEPA single or daily (4 fractions), 0.1–100 µM; IR daily, 4 fractions, 4 × 0.5–2.8 Gy (FaDu) or 4 × 1.6–8.2 Gy (A172); CIS, 0.1–30 µM (FaDu only); TMZ daily, 4 fractions, 5–100 µM (A172 only) and/or CCNU, 1–50 µM (A172 only). Seventy-two hours after the last treatment fraction, WST-1 assay was carried out according to manufacturers’ instructions and absorbance was measured at 435 nm with a reference wavelength of 680 nm on a spectrophotometer (SpectraMax i3x, Molecular Devices, LLC, San José, CA, USA) after thirty minutes.

ID_50_ and IC_50_ values were determined using exponential regression in Microsoft Excel 16.0 or, if not feasible, estimated directly from the graph of metabolic activity reduction (Table 1, Figure 1). For fractionated IR, experimentally determined ID_50_ values of both cell lines were averaged.

### 4.7. Cell Proliferation (EdU, BrdU)

Proliferation of CD34^+^ HSPCs was determined using the Click-iT^TM^ Plus 5-ethynyl-2’-deoxyuridine (EdU) Alexa Fluor^TM^ 488-Flow Cytometry Assay Kit (Invitrogen, Waltham, MA, USA) according to manufacturers’ instructions. Cells were seeded on a 96-well U-bottom plate (13,000/150 µL per well) and EdU (3.3 µM) was added. EdU assay was performed 24 h, 48 h, and 72 h after treatment with IEPA (1, 10, 100 µM) and IR (9 Gy). Washing steps were conducted using PBS containing 0.5% BSA and 2 mM/L EDTA. Preceding fixation, cells were additionally stained with APC mouse antihuman CD34 (1 µg/mL; clone 8G12; Becton Dickinson, Franklin Lakes, NJ, USA) for 30 min on ice and protected from light. At least 3000 cells were analyzed by flow cytometry (BD Accuri C6 Plus, Becton Dickinson, Franklin Lakes, NJ, USA).

Proliferation of tumor cells was investigated using 5-bromo-2′-deoxyuridine (BrdU) cell proliferation ELISA (Roche, Basel, Switzerland). Cells were seeded in a 96-well plate (FaDu: 2000/200 µL; A172: 2500/200 µL per well) in their respective culture medium and treated with IEPA (1, 10, 100 µM) and IR (9 Gy). BrdU assay was performed 48 and 72 h later according to manufacturers’ instructions. Twenty-four hours before assay execution, 10 µM BrdU was added. Absorbance was measured 30 min after adding the substrate solution using a microplate reader (SpectraMax i3x, Molecular Devices, LLC, San José, CA, USA) at 450 nm with a reference wavelength of 492 nm.

### 4.8. Apoptosis (Annexin-V)

Annexin-V assay was used in order to observe the influence of IEPA on apoptosis after treatment with IR or ChT agents in cancer cells and CD34^+^ HSPCs. Tumor cells were seeded in a 6-well plate (29,000–100,000 cells depending on mode of treatment) and treated with IEPA (1, 10, 20, 100 µM) and IR (single dose: 9, 15 Gy or fractionated: 4 × 1.8, 5.9 Gy). Seventy-two hours after single dose or last IR fraction, cells were harvested by trypsinization and washed once with PBS.

CD34^+^ HSCPs were seeded in a 96-well plate with at least 5000 cells per well and treated immediately with IEPA (1, 10, 100 µM), chemotherapeutics (1 µM CIS, 30 µM TMZ single dose, 60 µM CCNU), and IR (3.2 Gy, single dose). After 72 h, cells were washed once with PBS containing 0.5% BSA and 2 mM EDTA.

Staining with Annexin-V-FITC conjugate and PI (Annexin-V-FLUOS staining kit, Roche, Basel, Switzerland) was carried out according to manufacturers’ instructions after cell harvesting. Fluorescence was measured by flow cytometry (BD Accuri C6 Plus, Becton Dickinson, Franklin Lakes, NJ, USA) in at least 10,000 tumor cells and 3000 CD34^+^ HSCPs. Analysis of flow cytometry data was performed using dot plots visualizing both fluorescence intensities for each cell. For clustering of vital (Annexin-Vneg PIneg), early (Annexin-Vpos PIneg) and late apoptotic (Annexin-Vpos PIpos), and necrotic cells (Annexin-Vneg PIpos) four quadrants were de-fined. Examples of apoptosis induction after 15 Gy irradiation in A172 and FaDu cells are shown in Figure 3A.

### 4.9. Clonogenic Survival of Tumor Cells (Clonogenic Assay)

Clonogenic assay was conducted to examine long-term survival of tumor cell lines FaDu and A172. Cells were seeded in 6-well plates in duplicates of three different cell densities. Cells were treated with IEPA (10, 100 µmol/L), cytostatic agents (FaDu: 1 µM CIS; A172: 10 µM TMZ in 4 daily fractions, 10 µM TMZ single-dose, or 60 µM CCNU), or single-dose IR (3.2, 5.5, 9 Gy). After 7 days, 2 mL of culture medium was added. On days 14–16, cells were stained with Giemsa solution and consecutively scored as described previously [69].

### 4.10. Reactive Oxygen Species (DCFDA)

To determine intracellular ROS levels, cells were seeded in a 96-well cell culture plate (black, clear/flat bottom) at 10,000 cells per well for tumor cells and 15,000 cells for CD34^+^ HSPCs. The culture medium was replaced by PBS containing 10 µM 2′, 7′–dichlorofluorescein diacetate (DCFDA, Sigma-Aldrich, St. Louis, MO, USA). Cells were subsequently treated with IEPA (1, 10, 100 µM) and CIS (1 µM), TMZ (40 µM), CCNU (60 µM), or single-dose IR (5.5 Gy). A known radical scavenger, 2-Mercaptoethanol (β-ME; 5 mM; Sigma-Aldrich, St. Louis, MO, USA), was used as ROS inhibition control. Dichlorofluorescein (DCF) fluorescence measurement was conducted for up to 5 h beginning immediately after treatment using a microplate reader (Exc/Em = 495/529 nm, SpectraMax i3x, Molecular Devices, LLC, San José, CA, USA).

Additionally, in one representative experiment, the long-term intracellular ROS levels were measured 48 h after IR exposure or CIS treatment in tumor cells. Therefore, cells were seeded at 25,000 cells/well in a 24-well cell culture plate and treated with IEPA (10, 100 µM) and IR (5.5 Gy) or CIS (20 µM). Forty-eight hours later, cells were washed twice with PBS, harvested by trypsinization, and washed again with PBS (150× *g*, 5 min, 20 °C). The cell pellet was resuspended in PBS containing 10 µM DCFDA and incubated for 15–30 min at room temperature protected from light. After centrifugation (150× *g*, 5 min, 20 °C), pellet was resuspended in PBS and fluorescence was measured by flow cytometry (BD Accuri C6 Plus, Becton Dickinson, Franklin Lakes, NJ, USA) at channel FL1 (Exc/Em = 488/535 nm).

### 4.11. Cytokine Release (CBA)

The release of cytokines IL-1β, IL-6, IL-8, and TNF-α after treatment with IEPA and IR was investigated in both tumor cell lines, FaDu and A172, and in CD34^+^ HSCPs. Supernatants were collected 24, 48, and 72 h after treatment with IEPA, chemotherapeutic agents, or IR and stored at −70 °C. Cytometric Bead Array (CBA) Human Enhanced Sensitivity Master Buffer Kit and Enhanced Sensitivity Flex Sets (Becton Dickinson, Franklin Lakes, NJ, USA) for above-mentioned cytokines were used in a multiplex analysis according to manufacturers’ guidelines.

Fluorescence of capture beads and detection reagent was measured by flow cytometry (BD Accuri C6 Plus, Becton Dickinson, Franklin Lakes, NJ, USA). Obtained data were processed using FCAP Array software (v3.0.1, Beckton Dickinson, Franklin Lakes, NJ, USA) to determine the cytokine concentrations quantitatively.

### 4.12. Differentiation of CD34^+^ HSCPs (CFU Assay)

To evaluate treatment effects on the differentiation capacity and long-term survival of CD34^+^ HSPCs, a CFU cell assay was performed in semisolid methylcellulose-based MethoCult™ Medium H4434 Classic (Stemcell Technologies, Vancouver, BC, Canada). CD34^+^ cells were seeded in SFEM on a 48-well plate, treated with IEPA (1, 10, 100 µM), and irradiated with 3.2 Gy (single-dose) or 4 × 1 Gy (fractionated IR). Cells were then transferred to the MethoCult™ Medium containing respective concentrations of IEPA and the cytostatic agent (1 µM CIS, 40 µM TMZ, 60 µM CCNU) and seeded in duplicates on 6-well plates (300–1000 cells per well, adapted to treatment). At day 15/16, colonies were scored as BFU-E (burst-forming unit erythroid), CFU-G (colony-forming unit granulocyte), CFU-M (monocyte), CFU-GM (granulocyte-monocyte), or CFU-GEMM (granulocyte-erythroid-monocyte-megakaryocyte), as demonstrated in Figure 7B, using an inverted microscope (Axiovert 25, Carl Zeiss AG, Oberkochen, Germany) with a blue filter. During the counting period, culture plates were stored at 33 °C and 5% CO_2_ to inhibit growth.

### 4.13. DNA Double-Strand Breaks by γH2AX Assay

For quantification of radiation-induced DSBs, nuclear staining of the DNA repair protein γH2AX was performed. CD34^+^ HSPCs were seeded in a 96-well U-bottom cell culture plate (10,000 cells per well). Following treatment with IEPA (1, 10, 100 µM), cells were irradiated with 3.2 Gy or 5 Gy. After 30 min, 4 h, and 24 h, cells were stained for γH2AX as previously described [70]. ΓH2AX foci were counted to a maximum of 30 foci per nucleus in at least 30 nuclei per treatment group using fluorescence microscopy (Axiolab, Carl Zeiss AG, Oberkochen, Germany).

### 4.14. Statistics

Statistical analysis was performed using IBM SPSS 25 software. Homogeneity of variance was tested with Levene’s test based on the median before analysis of variance (ANOVA). Gabriel’s post hoc test (unequal sample sizes) and Tukey’s post hoc test (equal sample sizes) were chosen for further analysis. *p*-values ≤ 0.05 were considered as statistically significant (*; #) and *p*-values ≤ 0.01 (**; ##) and ≤ 0.001 (***; ###) as highly statistically significant.

To evaluate the effect of IEPA alone and in combination with RT and ChT on metabolic activity, a joint analysis of all IEPA-treated groups of both cell lines was conducted using three-way ANOVA with experiment number being assigned as random factor to compare the main effect of treatment with IEPA (independent variable) on absorbance (dependent variable).

## 5. Conclusions

Our data on HNSCC and glioblastoma tumor cells suggest IEPA to be a safe, promising candidate for investigation in clinical trials in a radio-oncological setting, as IEPA showed no tumor-protecting activities. With the use of compounds such as IEPA, more patients could potentially tolerate higher cumulative doses of chemotherapy; this appears important for reaching long-term survival, e.g., in cases of head and neck cancer.

We did not find mitigating effects due to IEPA on radiation- or chemotherapy-induced short- or long-term tumor cell death, although IEPA scavenged therapy-induced ROS. In addition, IEPA had no impact on RT-induced immune-modulating cytokine production.

Our investigations on CD34^+^ HSPCs demonstrated an IEPA-induced increase in hematopoietic progenitor cells, indicating preventive activity. However, the in vitro setting is not able to simulate BM microenvironment features. In vivo investigations of hematopoietic stem cells isolated from treated BM might further illuminate the acting mechanism of IEPA in these cells.

## Figures and Tables

**Figure 1 molecules-28-02008-f001:**
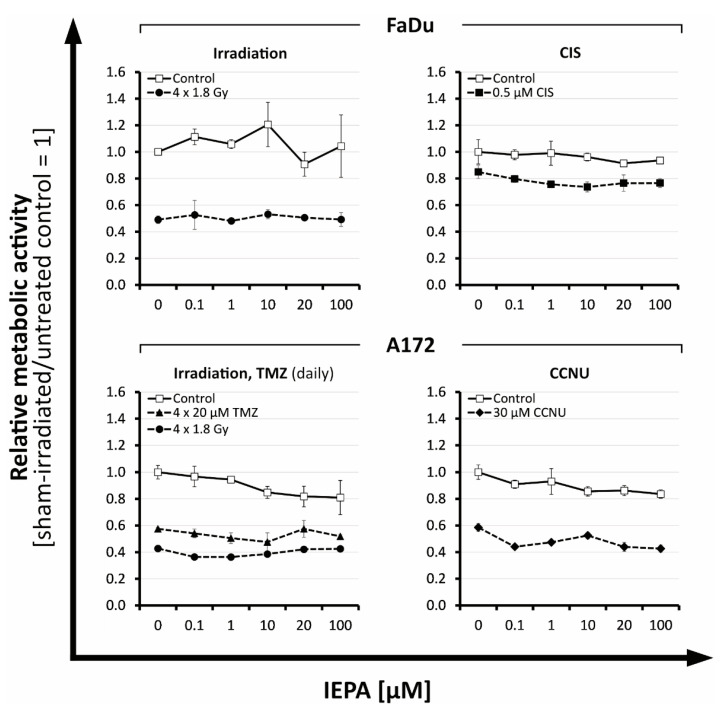
Relative effects of IEPA compared with sham-treated control on the metabolic activity of tumor cell lines FaDu and A172 after fractionated IR or treatment with different cytostatic agents at ID_50_/IC_50_. Data represent single experiments, except for fractionated control in A172 (n = 2). Experiments were performed in duplicates; mean ± SEM.

**Figure 2 molecules-28-02008-f002:**
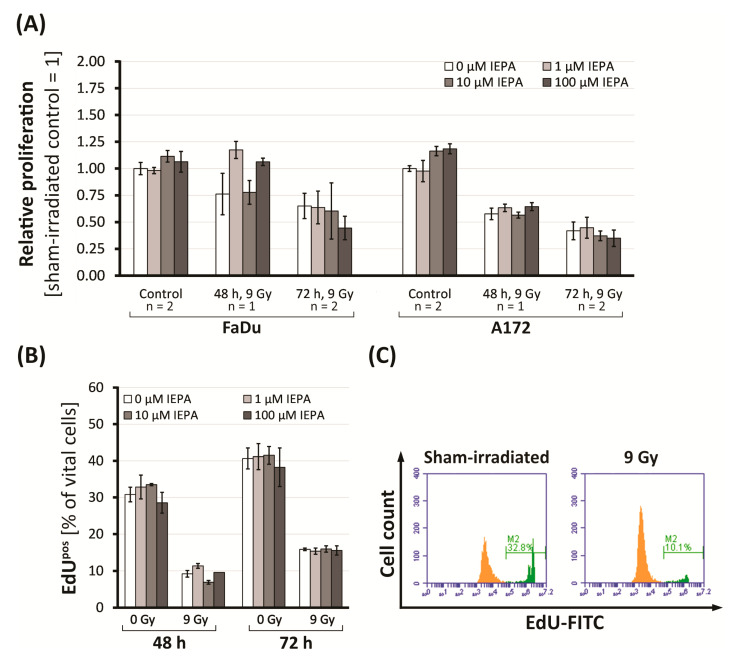
(**A**) Proliferation (BrdU assay) in FaDu and A172 cells 48 and 72 h after IR and treatment with IEPA related to control. For control and 72 h, mean values of two experiments ± SEM are presented. For 48 h, mean values of single experiments ± SEM conducted in triplets (FaDu) or quadruplets (A172) are shown. (**B**) Proliferation (% of EdU^pos^ cells) in CD34^+^ HSPCs after IR and treatment with IEPA. Data represent a single experiment in duplicates; mean ± SEM. (**C**) Flow-cytometric histogram of CD34^+^ HSPCs 48 h post-IR with fraction of EdU^pos^ cells highlighted in green.

**Figure 3 molecules-28-02008-f003:**
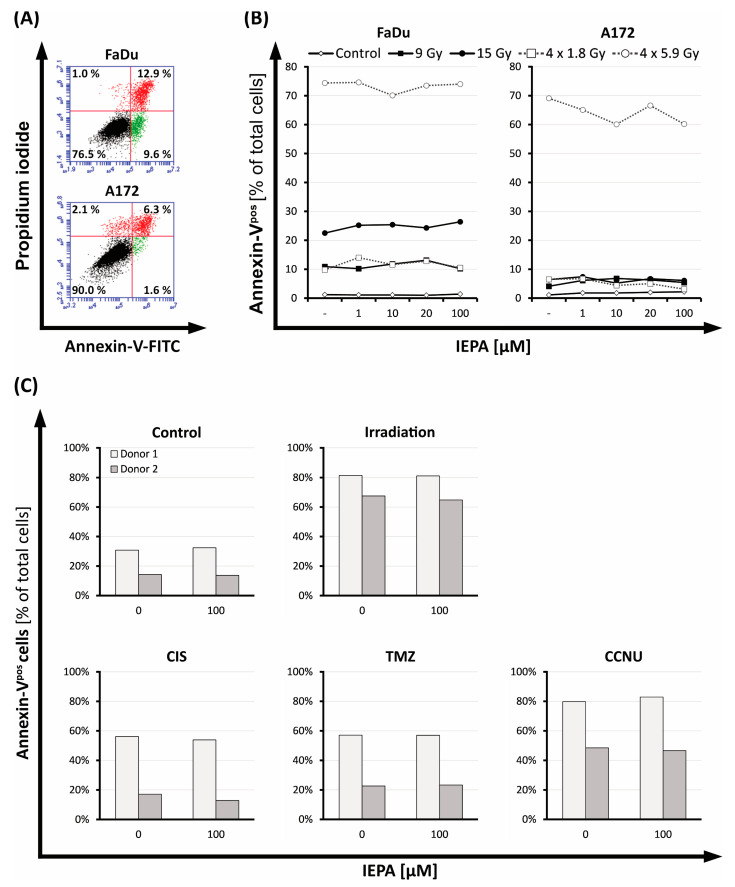
(**A**) Flow-cytometric dot plot of FaDu and A172 cells 72 h after IR (15 Gy), partitioned into vital cells (Annexin-V^neg^ PI^neg^, lower left quadrant), apoptotic cells (early apoptotic: Annexin-V^pos^ PI^neg^, lower right quadrant; late apoptotic: Annexin-V^pos^ PI^pos^, upper right quadrant), and necrotic cells (Annexin-V^neg^ PI^pos^, upper left quadrant). (**B**) Fraction of apoptotic cells (Annexin-V^pos^) in FaDu and A172 cells 72 h after IR in various regimens (single-dose: 9, 15 Gy; fractionated: 4 × 1.8, 4 × 5.9 Gy); treatment with IEPA compared with representative untreated control (sham-RT). (**C**) Fraction of apoptotic cells (Annexin-V^pos^) in CD34^+^ HSPCs 72 h after IR (3.2 Gy, single-dose), treatment with CIS (1 µM), TMZ (40 µM), or CCNU (60 µM), and application of IEPA (100 µM) compared with sham treatment.

**Figure 4 molecules-28-02008-f004:**
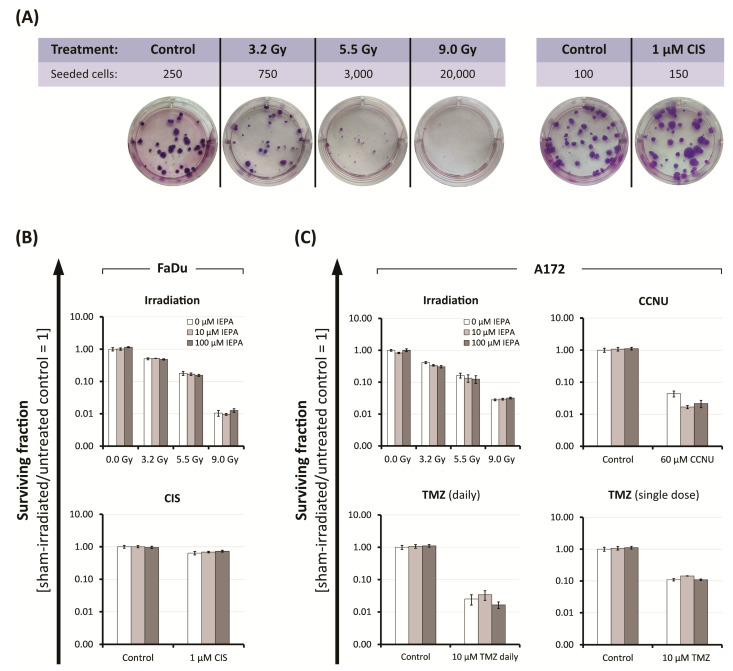
(**A**) Representative images of FaDu colonies, grown in a 6-well plate for clonogenic assay after IR and CIS treatment, stained with Giemsa. Colonies were scored 14-16 days after start of treatment. Clonogenic survival after treatment with IEPA of (**B**) FaDu cells irradiated or treated with CIS, and of (**C**) A172 cells irradiated or treated with CCNU, TMZ daily, or TMZ single-dose. Surviving fractions from one representative experiment conducted in three cell densities, each in duplicates, are presented as mean ± SEM.

**Figure 5 molecules-28-02008-f005:**
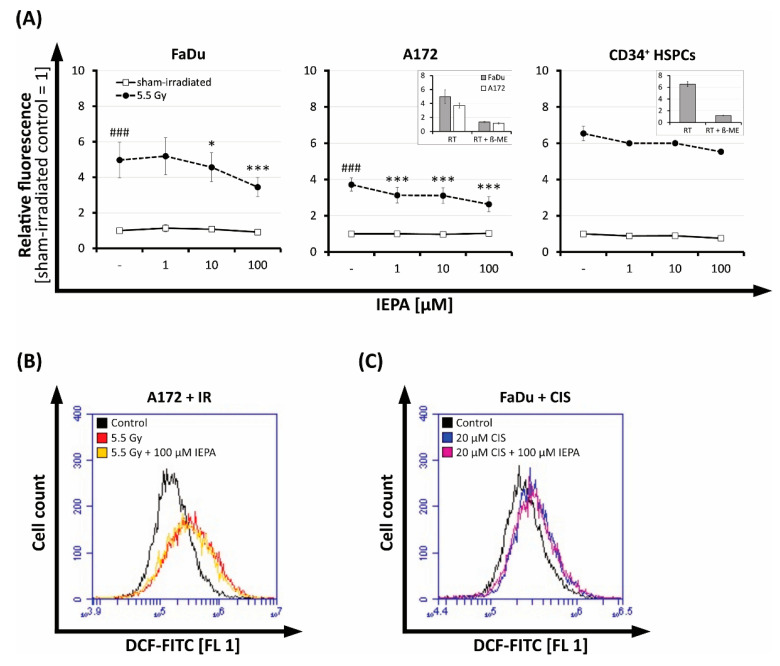
(**A**) Effect of IEPA (1, 10, 100 µM) after IR (5.5 Gy) on intracellular ROS levels indicated by DCF fluorescence in FaDu and A172 cells (mean ± SEM of three experiments in triplicates or quadruplicates) as well as CD34^+^ HSPCs (mean ± SEM of one experiment in duplicates). Statistical significance obtained from analysis of variance (ANOVA) and Gabriel post hoc test displayed as asterisks (***) compared with irradiated control and as hashes (###) compared with sham-irradiated control. Inserts depict corresponding inhibition controls with 5 mM ß-mercaptoethanol (β-ME). *, *p* ≤ 0.05. Flow-cytometric histograms of IEPA (100 µM) effects compared with untreated control on intracellular ROS 48 h after (**B**) IR (5.5 Gy) on A172 cells, or (**C**) CIS (20 µM) on FaDu cells.

**Figure 6 molecules-28-02008-f006:**
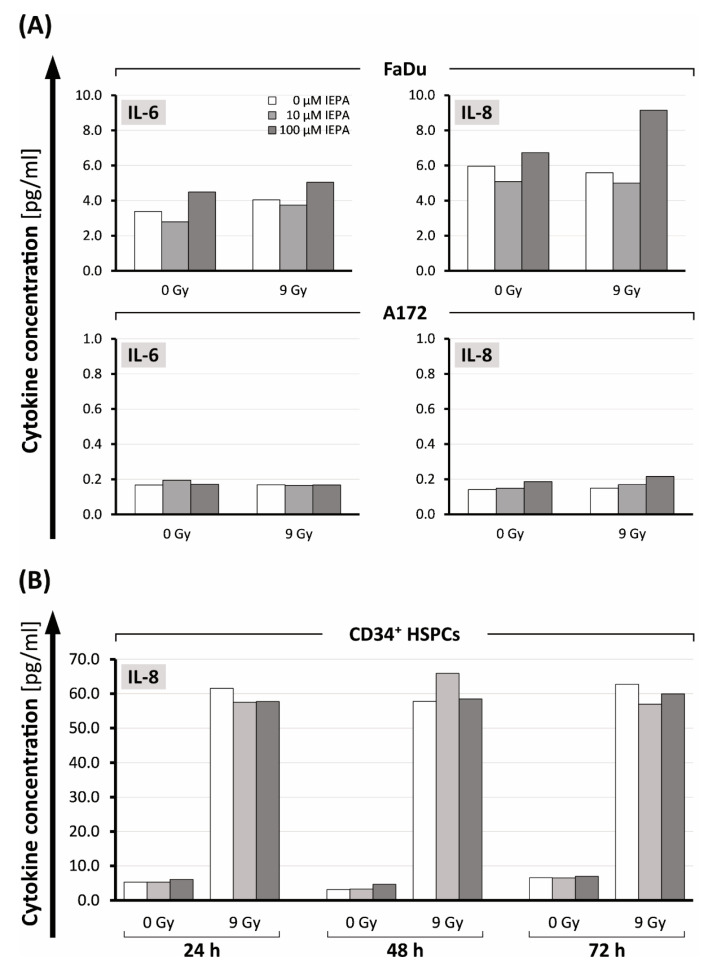
(**A**) IL-6 and IL-8 release 24 h after treatment with IEPA (10, 100 µM) as well as IR (9 Gy) in FaDu cells and A172 cells. (**B**) IL-8 levels in CD34^+^ HSCPs 24, 48, 72 h after IEPA application (10, 100 µM) and IR (9 Gy). All analyte concentrations obtained using flow-cytometric bead array (CBA) in one experiment with single measurements.

**Figure 7 molecules-28-02008-f007:**
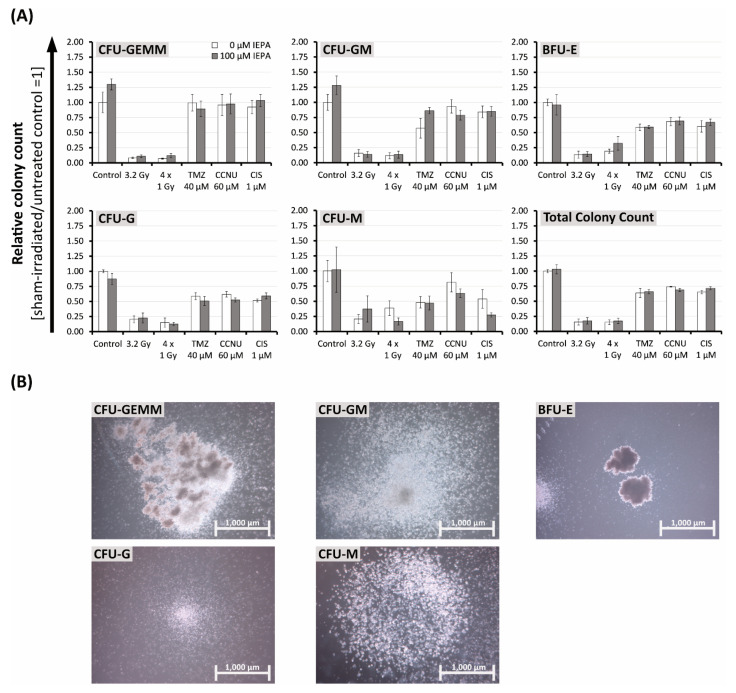
(**A**) Effects of IEPA (100 µM) combined with IR (single-dose: 5.5 Gy and fractionated: 4 × 1 Gy) or treatment with TMZ (40 µM, single-dose), CCNU (60 µM), or CIS (1 µM) on proliferation and differentiation capacity of CD34^+^ HSPCs in semisolid methylcellulose-based medium. Colonies scored after 14–16 days as CFU-GEMM (granulocyte-erythroid-monocyte-megakaryocyte), CFU-GM (granulocyte-monocyte), erythroid burst-forming units (BFU-E), CFU-G (granulocyte), and CFU-M (monocyte). Total colony count equals summation of all colony types. Data are presented as mean ± SEM of two experiments conducted in duplicates. (**B**) Depiction of the scored colony types appearing in our CFU cell assay.

**Figure 8 molecules-28-02008-f008:**
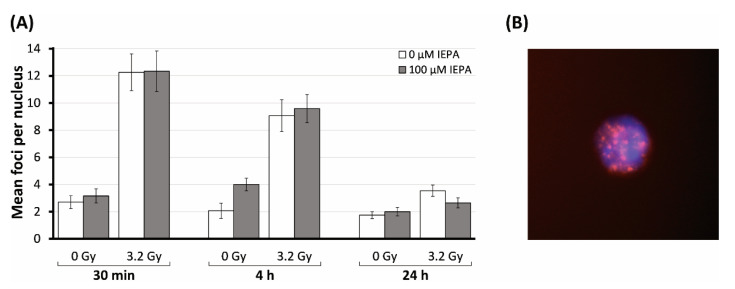
(**A**) Number of γH2AX foci in CD34^+^ HSPCs irradiated (3.2 Gy) and treated with IEPA (100 µM) over the course of 24 h compared with sham-irradiated control. Foci scored in 30 cells per group are presented as mean value ± SEM/nucleus. (**B**) Example of CD34^+^ HSPC in DAPI staining (blue) with IR-induced γH2AX foci (red); original magnification: ×100.

**Figure 9 molecules-28-02008-f009:**
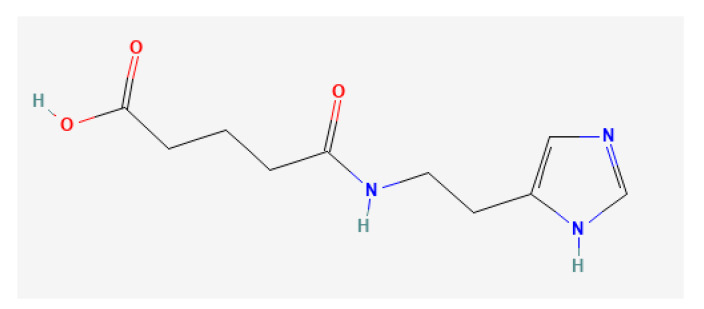
Chemical structure of IEPA (source: https://pubchem.ncbi.nlm.nih.gov/ (accessed on 6 February 2023)).

**Table 1 molecules-28-02008-t001:** Half-maximal inhibitory concentrations (IC_50_) for cytostatic agents (CIS, TMZ, CCNU) and doses (ID_50_) for IR (single-dose and fractionated) as determined by WST-1 assay in human tumor cell lines FaDu (HNSCC) and A172 (glioblastoma). For varying IR damage induction, multiples of ID_50_ were calculated from multiples of ID[BED]_50_ (see Section 4) using linear–quadratic model and are listed below.

IC_50_: Cytostatic Agents	FaDu		A 172	
CIS	0.5 µM		-	
TMZ daily	-		4 × 20 µM	
CCNU	-		30 µM	

**ID_50_: Irradiation**	**FaDu + A172 combined**			
ID_50_ (4 fractions) *Experimentally determined*	1.8 Gy/fraction			
↪ ID[BED]_50_ (biologically effective dose) *Calculated, α/β = 10*	8.5 Gy			
↪ ID_50_ (single dose) *Calculated*	5.5 Gy			

**Multiples of ID_50_**	**Single dose**	**BED**		**4 fractions**
0.25 × ID_50_	2.1 Gy	2.1 Gy		4 × 0.5 Gy
0.5 × ID_50_	3.2 Gy	4.3 Gy		4 × 1.0 Gy
1 × ID_50_	5.5 Gy	8.5 Gy		4 × 1.8 Gy
2 × ID_50_	9.0 Gy	17.1 Gy		4 × 3.2 Gy

## Data Availability

The data presented in this study are available on request from the corresponding author.

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
