# Peer review of "Imidazolyl Ethanamide Pentandioic Acid (IEPA) as Potential Radical Scavenger during Tumor Therapy in Human Hematopoietic Stem Cells"

_molecules, 2023, doi:10.3390/molecules28052008_

Round 1
Reviewer 1 Report
Dear Authors,
The theme that you chose for your work is very interesting, due to the urgent need for discovering new substances that may reduce the adverse effects of radiotherapy or/and chemotherapy, for the oncologic patients.
I have some corrections/suggestions in order to improve your manuscript:
1. I consider necessary to add also the chemical structure of imidazolyl ethanamide pentandioic acid
2. I suggest you make more discussions regarding the mechanism of action of IEPA and the structure-activity relationship for this compound. You present it like an antiviral used in Russia, so please make a better connection with the hematopoietic protective effect.
3. all authors seem to belong to the same department, so why are there different indices for them?
4. detail what FaDu and A172 in line 110 stand for
5. please detail the conclusions
6. I think that you wanted to write "safe", not "save" in line 612
7. I suggest to continue the investigation of IEPA effect on other tumor cell lines and also after other drugs used in chemotherapy
Author Response
Dear Reviewer,
thanks a lot for your time to evaluate our manuscript and giving suggestions to enhance the quality.
Answers to your comments
- I consider necessary to add also the chemical structure of imidazolyl ethanamide pentandioic acid
We have inserted the 2D structure of IEPA in figure 9.
- I suggest you make more discussions regarding the mechanism of action of IEPA and the structure-activity relationship for this compound. You present it like an antiviral used in Russia, so please make a better connection with the hematopoietic protective effect.
The mode of action of IEPA as hematopietic drug is not examined yet and is part of our studies.
We have modified the introduction to clarifiy the correlation between its treatment effects seen in viral infections and its cytoprotective activity (line 78/79). So, IEPA is not only an anti-virus treatment with the effect of killing the virus but rather protecting the infected cell from cell death. This might be the same molecular mechanism seen in hematopoietic cells but this is too much speculative as there are no available literature for this hypothesis.
- all authors seem to belong to the same department, so why are there different indices for them?
We have edited this. Thanks for this comment.
- detail what FaDu and A172 in line 110 stand for
We have add this.
- please detail the conclusions
We have detailed the conclusions.
- I think that you wanted to write "safe", not "save" in line 612
Thank you.
- I suggest to continue the investigation of IEPA effect on other tumor cell lines and also after other drugs used in chemotherapy
Thank you for this suggestion.

Reviewer 2 Report
The manuscript describes the investigation use of of IEPA together with different treatments including IR, ChT, CIS, CCNU, and TMZ to compare the toxicity after the treatment. However, the manuscript was not clearly written, the idea and purpose of the research was not explained clearly. Also the experimental designs and data are difficult to follow and understand. The data presentation must be improved. Thus, I believe at this stage, the manuscript is not ready to be published in the journal of Molecules. Please see the comments below.
1. Conclusion, line 1. "to be a save, promising..." should be "to be a safe and promising..."
2. Figure 2A and Figure 4, the error bars are too thin to be visible.
3. For IL-6 and IL-8 release experiment, how many repeats did authors performed? Should there be error bars and P value?
4. The comparison in Figure 1, can authors explain the different of dosage in each experiment? Why TMZ were treated daily while CCNU is 30 uM once? what is the rationale?
5. The experimental reagents (e.g. DCFDA, Annexin-V)should not be listed in the abstract.
6. In Figure 1, caption, (=1) do you mean n=1? if n=1 why there are error bars and mean SEM?
7. Figure 2A and 2B, the colours of the bar chart are too similar.
Author Response
Dear Reviewer,
thanks a lot for your time to evaluate our manuscript and giving suggestions to enhance the quality.
We tried to refine and strengenth the idea and purpose of our research in the introduction section (line 64-67 and 95/96)
We improve the description of experimental design and include a sentence within the result section to explain the different dosing regimes (line 112-114).
The data presentation was improved by editing figures and legends.
Answers to your comments
- Conclusion, line 1. "to be a save, promising..." should be "to be a safe and promising..."
Thanks a lot for this hint.
- Figure 2A and Figure 4, the error bars are too thin to be visible.
The figures were edited.
- For IL-6 and IL-8 release experiment, how many repeats did authors performed? Should there be error bars and P value?
For the cytokine release experiments, only one experiment as single measurement was performed in the two cell lines at different time points. Therefore, no statistical analyses was possible (see legend Figure 6).
- The comparison in Figure 1, can authors explain the different of dosage in each experiment? Why TMZ were treated daily while CCNU is 30 uM once? what is the rationale?
Thank you for giving this note. The different dosage schedules were adapted from the clinical setting where TMZ is given daily and CCNU once a week to the tumor patients. We mentioned this within the method part (line 466). In addition, we now implemented a notice at the beginning of the results part (line 112-114).
- The experimental reagents (e.g. DCFDA, Annexin-V)should not be listed in the abstract.
We have removed the reagent names.
- In Figure 1, caption, (=1) do you mean n=1? if n=1 why there are error bars and mean SEM?
Thank you for giving this note. It seems to get confusions between n = 1 and (=1). (=1) in this context ment that we depicted the values compared to control which is 1 (=1). We have removed (=1) in the legend, because it should be clear as the y axis is labeled [sham-irradiated/untreated control =1] precise enough.
There are error bars as one experiment per treatment condition was performed in duplicates (see figure legend). Thereout, mean and SEM are calculated and depicted.
- Figure 2A and 2B, the colours of the bar chart are too similar.
The figures were edited.

Round 2
Reviewer 2 Report
After the improvement was made, the manuscript is now easier to read and to understand. There are one minor things would like to comment before accepting.
Figure 3A was not mentioned and explain in the text. Could authors explain a bit about it? Also the percentage number of the late apoptotic quadrant was not clear. It is masked by the red colour cell and it maybe too small and blur.
Author Response
Thank you for this valuable hint.
We have improved the figure 3A for better visibility and integrated a passage in method section 4.8 (line 529) to explain the figure.